# Phenology of Vegetation in Arid Northwest China Based on Sun-Induced Chlorophyll Fluorescence

**Zhizhong Chen [1,†], Mei Zan [1,2,*,†], Jingjing Kong [1,2], Shunfa Yang [1,2] and Cong Xue [1,2]**

[1] School of Geographical and Tourism, Xinjiang Normal University, Urumqi 830054, China; 107622021210516@stu.xjnu.edu.cn (Z.C.); 107622021210529@stu.xjnu.edu.cn (J.K.); 107622022210578@stu.xjnu.edu.cn (S.Y.); xuecong@stu.xjnu.edu.cn (C.X.)

[2] Xinjiang Key Laboratory of Lake Environment and Resources in Arid Zone, Urumqi 830054, China

\* Correspondence: 107622007010058@xjnu.edu.cn; Tel.: +86-135-6590-9659

† These authors contributed equally to this work.

**Abstract:** The accurate monitoring of vegetation phenology is critical for carbon sequestration and sink enhancement. Vegetation phenology in arid zones is more sensitive to climate responses; therefore, it is important to conduct research on phenology in arid zones in response to global climate change. This study compared the applicability of the enhanced vegetation index (EVI), which is superior in arid zones, and global solar-induced chlorophyll fluorescence (GOSIF), which has a high spatial resolution, in extracting vegetation phenology in arid zones, and explored the mechanism of the differences in the effects of environmental factors on the phenology of different vegetation types. Therefore, this study employed a global solar-induced chlorophyll fluorescence (GOSIF) dataset to determine the start and end of the vegetation growth season ($SOS_{SIF}$ and $EOS_{SIF}$, respectively) in the arid zone of Northwest China from 2001 to 2019. The results were compared with those from the EVI-based MODIS climate product MCD12Q2 ($SOS_{EVI}$ and $EOS_{EVI}$). Variations in the sensitivity of these climatic datasets concerning temperature, precipitation, and standardised precipitation evapotranspiration index (SPEI) were assessed through partial correlation analysis. Results: Compared to the MCD12Q2 climatic products, $SOS_{SIF}$ and $EOS_{SIF}$ closely matched the observed climate data in the study area. Spring onset was delayed at higher altitudes and latitudes, and the end of the growing season occurred earlier in these areas. Both $SOS_{SIF}$ and $EOS_{SIF}$ significantly advanced from 2001 to 2019 (trend degrees −0.22 and −0.48, respectively). Spring vegetation phenology was chiefly influenced by precipitation while autumn vegetation phenology was driven by both precipitation and SPEI. GOSIF-based climate data provides a more accurate representation of vegetation phenology compared to traditional vegetation indices. The findings of this study contribute to a deeper understanding of the potential ability of EVI and SIF to reveal the influence of vegetation phenology on the carbon cycle.

**Keywords:** vegetation phenology; GOSIF; partial correlation analysis; SPEI; arid zone

## 1. Introduction

Global change research has primarily focused on assessing the impact and feedback loops of climate change on terrestrial ecosystems [1]. The growth and transformation of vegetation are pivotal in shaping the structure and functions of ecosystems and influencing aspects that include carbon cycling, photosynthesis, and species composition. These changes, in turn, have broader impacts on the climate system [2]. It is important to recognise that different vegetation types exhibit varying responses to climate change, and their spatial and temporal distributions differ significantly. This underscores the significance of vegetation phenology, a field dedicated to studying the interplay between plants (including crops) and their environment. The goal is to uncover the patterns and mechanisms governing shifts in the physiological cycles of vegetation. This knowledge is crucial both for agricultural production and scientific research [3].





The United Nations Intergovernmental Panel on Climate Change (IPCC) has highlighted the importance of vegetation phenology as a sensitive indicator of terrestrial ecosystem responses to climate change [4,5]. Consequently, monitoring vegetation phenology and studying the cyclical patterns of change hold significant theoretical and practical value.

The recent availability of satellite remote sensing data has become a valuable tool for monitoring vegetation phenology across different ecosystems and scales [6,7]. Traditional remote sensing methods for climate estimation mainly include a vegetation index (VI) derived from reflectance data to assess vegetation greenness and invert vegetation climate [8,9], such as the normalized vegetation index (NDVI) and enhanced vegetation index (EVI), and remote sensing indices reflecting vegetation canopy. Other remote sensing indices that reflect the vegetation canopy and weaken the influence of the background, such as the plant phenology index (PPI), which reduces the influence of soil, rain, and snow; the normalised differential green index (NDGI), which reduces the influence of fall foliage; the enhanced vegetation index (EVI), which is insensitive to soil brightness; differential green index (NDGI), which reduces the effects of fall foliage; and vegetation phenology and productivity (VPP). However, previous studies have found that PPI have greater limitations in arid regions [10]. NDGI has better accuracy for monitoring the phenology of grasslands with snow cover, whereas vegetation phenology without snow cover is subject to greater uncertainty [11]. The higher spatial resolution of VPP data is also suitable for phenological research, but the VPP dataset that can be directly accessed at present has only been available since 2017; therefore, the use of VPP for the long time series of phenological studies is still limited [12]. Some studies have shown that EVI is more suitable for vegetation phenology research in arid zones than other vegetation indices that characterise canopy morphology [13,14]. Moreover, traditional vegetation indices are limited due to their correlation with the morphological characteristics of vegetation canopies, resulting in errors when characterising vegetation phenology based solely on canopy morphology, as observed in the study by Miao et al. (2017) [15]. Gross primary productivity (GPP), the amount of organic carbon fixed by photosynthesis, is the most directly relevant factor for vegetation phenology, but it is commonly used as a validation dataset as it contains flux site data, which are spatially discontinuous and difficult for large-scale plant phenology monitoring [16,17]. SIF is an indicator of the photosynthetic performance of vegetation that can accurately reflect vegetation phenology [18,19]. Therefore, in this study, remote sensing indices suitable for different vegetation types in arid zones were selected from different angles to carry out long time series climate studies.

Sunlight-induced chlorophyll fluorescence (SIF) is a type of fluorescence emitted by vegetative chloroplasts that absorb active radiation photosynthetically at a wavelength of 650–800 nm under sunlight [18]. In contrast to the vegetation index of canopy reflectance, SIF is closely related to photosynthesis, is less affected by cloud cover and atmospheric scattering, and is a fast, direct, and non-invasive indicator of vegetation photosynthetic performance. Moreover, the cyclic change rule of SIF is more obvious [19] and can provide a favourable basis for research on vegetation phenology, vegetation drought stress, disease and pest monitoring, vegetation yield estimation, and the carbon cycle [20,21]. Some studies have shown that the SIF of forests at high latitudes is more sensitive to climate change than traditional vegetation indices [22]. The current remote sensing satellites for acquiring SIF are mainly the Carbon Observatory-2 (OCO-2) and Global Ozone Monitoring Experiment-2 (GOME-2); however, the spatial resolution of the acquired SIFs is low, which brings a great deal of uncertainty to the remote sensing estimation of vegetation phenology [23]. Therefore, the inversion of vegetation phenology using high spatial and temporal resolution SIF data has important practical application value.

Currently, the methods of vegetation phenology extraction through vegetation indices mainly include two categories, in which rule-based phenology extraction methods (e.g., amplitude threshold, first-order derivative, second-order inverse, third-order derivative, relative change curvature, and rate of change in curvature) have been widely used for vegetation phenology extraction due to their simplicity, ability to minimise the influence

of background interference, and lack of a need for a variety of sample data; however, the accuracy greatly depends on the selection of thresholds [24,25]. The other category is machine-learning-based climate extraction methods (e.g., random forest and neural network models), which are more accurate in extracting vegetation phenology from remotely sensed data; however, their accuracy is limited by the number and quality of training samples as they require a large number of ground-based climatic observations as samples for training models [24,26]. All the above methods can be used to extract vegetation phenology information from coarser resolution remote sensing indices.

While extensive research has explored vegetation phenology in humid climates, there is a distinct need to address this issue in arid regions, which encompass a substantial 41% of the global land area. Vegetation in arid regions demonstrates heightened sensitivity to climate change, yet their responses on a global scale remain unclear [27,28]. Arid regions in Northwest China exhibit considerable variability in topography, geomorphology, and climate, resulting in substantial spatial and temporal variations in vegetation phenology [29]. Understanding the mechanisms governing plant phenology responses to climate change in arid and semi-arid regions of China is vital for broader studies on vegetation and global change. This knowledge also serves as a theoretical foundation for adapting to global change [30,31].

The vegetation in arid northwest China was selected as the research subject in this study. The latest high-resolution global solar-induced chlorophyll fluorescence (GOSIF) dataset from 2001 to 2019 was utilized, and the Savitzky–Golay (S–G) and dynamic threshold methods were employed for estimating the phenological characteristics of vegetation. A comparative analysis with the MODIS phenological product MCD12Q2 was conducted, followed by a discussion on the responses of two climate datasets to various climate factors through partial correlation analysis. The research objectives of this study were to (1) identify more suitable vegetation indices for the inversion of vegetation phenology in arid zones, (2) clarify the effects of environmental factors (temperature, precipitation, and SPEI) on the phenology of different vegetation types, and (3) explore the reasons for the differences in the sensitivity of vegetation phenology to climatic factors. The results of this study provide a theoretical basis for the response of vegetation phenology to climate change and the study of the relationship between vegetation phenology and the carbon cycle in the northwest arid zone.

## 2. Materials and Methods

### 2.1. General Description of Study Area

The arid region of Northwest China is located in the heart of the Eurasian continent at mid-latitudes. Precipitation primarily originates from mountainous areas and glacial meltwater, resulting in an average annual precipitation of <200 mm, characteristic of a continental climate [32]. Geographically, the region ($34°44'11''–49°077'24''$ N, $73°47'17''–106°43'00''$ E) spans from the Pamir Plateau in the west to the Helan Mountains in the east, and from the southern periphery of the Tarim Basin in the south to the Altai Mountains in the north. It encompasses a wide range of altitudes, extending from 192 to 8545 m, covering approximately $2.6 \times 10^6$ km$^2$, equivalent to 30% of the total land area of China [33]. Within this expanse, there are areas designated for forest land, grassland, cropland, and other land-use types, measuring $6.6 \times 10^4$, $5.1 \times 10^4$, $4.1 \times 10^4$, and $6.7 \times 10^3$ km$^2$, respectively [33]. The study area encompasses the Xinjiang Uyghur Autonomous Region, Alashan Plateau of Inner Mongolia, and the area west of the Helan Mountains in the Ningxia Hui Autonomous Region (Figure 1). This area is a vital component of the Central Asian arid zone, ranking among the world's driest regions at its latitude [34]. Its delicate ecological environment is highly susceptible to global changes [35]. Therefore, it is imperative to investigate the influence of climate change on vegetation growth within this region to preserve ecosystem stability.

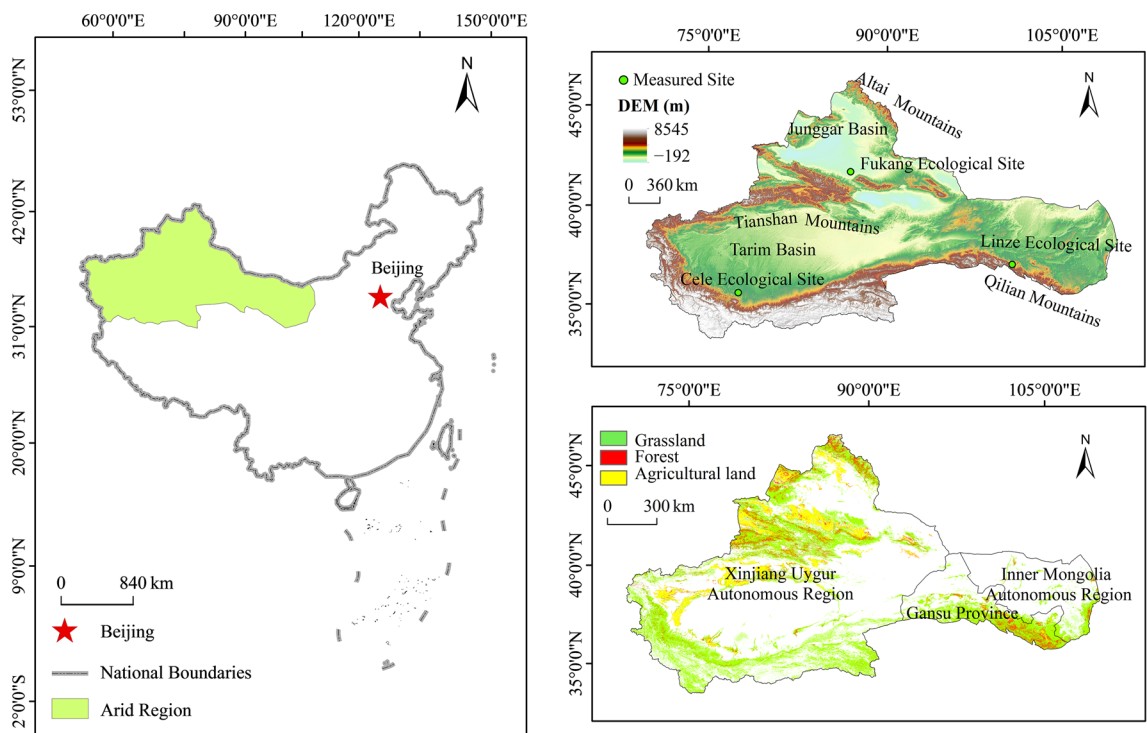

**Figure 1.** Overview map of the study area.

*2.2. Data Sources and Preprocessing*

2.2.1. SIF, EVI, and Gross Primary Production (GPP) Data

Global SIF product (GOSIF) based on Orbiting Carbon Observatory-2 (OCO-2) data was utilised. The spatial and temporal resolution of the dataset from Global Ecology Group's data repository was 0.05° and 8 d, respectively. (available at http://data.globalecology.unh.edu/data/GOSIF_v2/Annual/, accessed on 6 May 2023), and was created by integrating discrete OCO-2 SIF point clouds with remote sensing data from MODIS and meteorological reanalysis data, spanning from 2001 to 2019.

To ensure the accuracy of our vegetation phenology estimation, we used ArcGIS10.8 software to eliminate data rasters with pixel values of 32,767, representing water bodies, and values of 32,766, indicating year-round snow and ice. This step aimed to mitigate potential interference from water bodies, snow, and ice on our vegetation phenology assessments [36].

For comparative analysis, we obtained EVI and GPP data from the MOD13A1 and MOD17A2H.061 products of MODIS data, accessible through NASA in the United States (https://lpdaac.usgs.gov/, accessed on 5 May 2023). These datasets have temporal resolutions of 16 and 8 days, respectively, with a spatial resolution of 500 m.

To ensure consistency in our analysis, we employed the maximum value synthesis method of ArcGIS10.8 software to harmonise the temporal resolutions of SIF, GPP, and EVI in the study area from 2001 to 2019 at intervals of 16 days. Additionally, we aligned the spatial resolutions of the EVI and GPP datasets with those of SIF, setting them at 0.05° using a downscaling method [37].

2.2.2. Vegetation Type and Meteorological Data

Vegetation type information obtained from the Northwest Institute of Eco-Environment and Resources combined with data from the Second Land Survey (accessible at http://www.nieer.cas.cn/kyfw/kxsj/, accessed on 20 May 2023) were extracted to create a comprehensive vegetation type map of the study area (Figure 1). The prominent vegetation types within the arid zone of Northwest China encompass woodland, grassland, and cropland.

We sourced 1 km month-by-month gridded temperature (TMP) and precipitation (PRE) data for the study area from 2001 to 2019 through the National Earth System Science Data Center (available at http://gre.geodata.cn, accessed on 1 June 2023). This dataset is a fusion of global 0.5° climate data from the Climatic Research Unit, University of East Anglia, and high-resolution climate data from WorldClim (https://worldclim.org, accessed on 2 June 2023), achieved via spatial downscaling methods. Importantly, it has undergone rigorous validation with data obtained from 496 independent meteorological observation sites [38].

To characterise drought conditions within our study area from 2001 to 2019, we employed the standardised precipitation evapotranspiration index (SPEI) with a spatial resolution of 0.5°, which were obtained from the global SPEI database SPEI base v2.7. (https://digital.csic.es/handle/10261/268088, accessed on 2 June 2023). The SPEI amalgamates the strengths of both the Palmer drought severity index (PDSI) and the standardised precipitation index (SPI). SPEI accounts for the processes of moisture and heat balance, offering insights into the extent of the surface water deficit and its accumulation. The China Meteorological Administration (CMA) provides the SPEI drought classification criteria, with SPEI categories as follows: $SPEI > -0.5$ indicates no drought; $-1 < SPEI \leq -0.5$ signifies mild drought; $-1.5 < SPEI \leq -1$ denotes moderate drought; $-2 < SPEI \leq -1.5$ corresponds to severe drought; and $SPEI \leq -2$ represents exceptional drought (https://www.cma.gov.cn/, accessed on 3 June 2023).

### 2.2.3. MODIS Climate Data and Ground-Based Climate Observations

For the comparative analysis, MODIS phenology product data, MCD12Q2, from the NASA website (https://lpdaac.usgs.gov/, accessed on 4 June 2023) were obtained. These data have a spatial resolution of 500 m. Greenup in the MCD12Q2 product is the period of onset of vegetative growth ($SOS_{EVI}$) and dormancy is the period of the end of vegetative growth ($EOS_{EVI}$). The MCD12Q2 product utilises a segmented logistic function fitted to the EVI from MODIS as the primary data source. It identifies the extremes of curvature change points in the EVI to determine the initiation, maturity, peak, and conclusion of vegetation growth [37].

MCD12Q2 products spanning 2001 to 2019 were processed by splicing, projecting transformation, and cropping to extract data on the $SOS_{EVI}$ and $EOS_{EVI}$ of vegetation within our study area. This allowed calculation of the duration of the vegetation growth period. Climatic parameters were expressed as Julian days, converting the dates of climatic phenomena into the actual number of days from January 1 of the current year (days of the year, doy), thereby creating a time series for each climatic period.

To validate the accuracy of our findings, we sourced ground-based climatic observations from the National Ecological Science Data Center (http://rs.cern.ac.cn/data/meta?id=40177, accessed on 4 June 2023). These observations included climatic data for both herbaceous and woody plants at 21 ecological stations across China, covering the period from 2003 to 2015. To ensure precision, climate observation data from the Fukang, Celle, and Linze stations were selected, as they had recorded data for at least three years and monitored at least eight plant species annually. Additional information about these stations is presented in Table 1.

**Table 1.** Sites analysed in the study.

| Field Test Site Name | Longitude | Latitude |
|---|---|---|
| Linze Station | 99°35″ | 39°04′ |
| Fukang Station | 87°55′ | 44°17′ |
| Cele Station | 80°43′ | 37°00′ |

The Fukang, Celle, and Linze stations are strategically located in desert, grassland, and cropland areas, respectively, and are representative of typical vegetation ecosystems in the northwestern arid zone [36]. For woody plants, the start of the growth period was

defined as the leaf-spreading phase, and the end of the period was set as the deciduous phase. For herbaceous plants, the start of the growth season was defined as the average of the greening and flowering periods, while the end of the season was defined as the yellowing period. Since remote sensing technology relies on vegetation greenness data to calculate phenology parameters, there may be some discrepancies with the phenology periods observed at ground stations. To minimise errors, the averages of the different growth seasons of herbaceous and woody plants were defined as the beginning and end of the growing season for each year at the respective station. Data for this site are from the National Ecological Science Data Centre (NESDC) Resource Sharing Service Platform. (http://rs.cern.ac.cn/, accessed on 6 June 2023) [39].

*2.3. Vegetation Phenology Extraction*

2.3.1. Time Series Reconstruction of SIF Data

SIF time series data are robust against cloud cover and atmospheric scattering. However, random factors and missing values can introduce noise. Therefore, data smoothing is a necessary step. In comparison to the double logistic (DL) and asymmetrical Gaussian (AG) filtering methods, the Savitzky–Golay (S–G) smoothing technique provides smoother and more stable reconstructed curves [40]. This method effectively reduces noise while preserving the nuances and fine details in long time series data [41].

In this study, we employed the S–G filter, which is available in the TIMESAT3.3 software, to reconstruct the SIF data time series from 2001 to 2019 within the study area. This filtering method yielded smoothed curves that can more accurately capture the peaks and valleys within the SIF time series data [20]. The S–G filter utilises an iterative algorithm that relies on convolutional fitting using the least squares method [42]. The process involves specific steps that are detailed below.

In the initial step, for the peaks and valleys evident in the SIF curves before denoising, the left and right segments of the curves are fitted using the S–G function. The local fitting function ($f1(t)$) is expressed by Equation (1):

$$f1(t) \equiv f(t; a1, a2, b1, \ldots, b5) = a1 + a2g(t; b1, \ldots, b5) \tag{1}$$

where a1 and a2 are linear parameters that ensure a reasonable rate of increase or decrease in the basis function span.

Step two is

$$g(t; \, b1, \ldots, b5) = \begin{cases} exp\left[-\left(\frac{1-b_1}{b_2}\right)^{b_3}\right] & (t > b1) \\ exp\left[-\left(\frac{1-b_3}{b_4}\right)^{b_5}\right] & (t > b2) \end{cases} \tag{2}$$

where $g(t; b1, \ldots, b5)$ is the S–G function, b1 is the location parameter of the peak or trough of the time variable *t*, and *b2*, *b3*, *b4*, and *b5* are the width and steepness of the left and right halves of the fitted curves, respectively.

The local fitting function model can be used to plot the maximum and minimum values of the SIF curve. The overall fitting function combines the characteristics of the local fitting function [43], which is calculated as

$$f2(t) = \begin{cases} \alpha(t)f_L(t) + [1 - \alpha(t)]f_C(t) & (t_L < t < t_C) \\ \beta(t)f_C(t) + [1 - \beta(t)]f_R(t) & (t_L < t < t_C) \end{cases} \tag{3}$$

where $f2(t)$ is the overall fitting function, $[t_L, t_C]$ is the variation interval of the SIF data series, $f_L(t)$, $f_C(t)$, and $f_R(t)$ represent local functions corresponding to the left valley, middle peak, and right valley in the interval $[t_L, t_C]$, respectively, and $\alpha(t)$ and $\beta(t)$ are the truncation functions located between [0, 1]. The merging of locally fitted functions is a key feature of

Gaussian function simulations. This approach increases the flexibility of the fitted functions to simulate more complex time series variations [44].

### 2.3.2. Dynamic Thresholding

Based on the reconstruction of SIF time series data from 2001 to 2019, the dynamic threshold method was used to extract the vegetation phenology parameters $SOS_{SIF}$ and $EOS_{SIF}$. First, the dynamic threshold of SIF ($SIF_{ratio}$) was calculated as

$$SIF_{ratio} = \frac{SIF_t - SIF_{min}}{SIF_{max} - SIF_{min}} \tag{4}$$

where $SIF_t$ is the SIF value for a given day in a given year, and $SIF_{max}$ and $SIF_{min}$ are the annual maximum and minimum SIF values, respectively, for the time span of the study. $SOS_{SIF}$ is defined as the doy when the SIF value exceeds the local threshold $SIF_{ratio}$. $EOS_{SIF}$ is defined as the doy when the SIF value is below the local threshold $SIF_{ratio}$. Considering the effects of different wavelengths of light on chlorophyll fluorescence, this threshold was set to one-fifth of the difference between $SIF_{max}$ and $SIF_{min}$ [45].

### 2.3.3. Sen and Mann–Kendall Trend Analyses

Sen trend degree was first used to analyse the trend of climate information. Then, Matlab R2021b software was used to test the significance of climate data trends by Mann–Kendall test [46]. The degree of Sen trend is calculated as follows:

$$\beta_{sif} = median\left(\frac{x_j - x_i}{j - i}\right), \ 1 < i < j < 19 \tag{5}$$

where $x_j$ and $x_i$ are the annual mean vegetation phenology doy for the $j$-th and $i$-th year image elements, respectively, $\beta sif$ is the slope of phenology change, $\beta_{sif} > 0$ indicates an upward trend in phenology data, and $\beta_{sif} < 0$ indicates a downward trend in phenology.

The $S$ statistic for the Mann–Kendall trend significance test was calculated as follows:

$$S = \sum_{i=1}^{n-1} \sum_{j=i+1}^{n} sgn(x_j - x_i) \tag{6}$$

where the $sgn$ function is a step function that symbolises the difference between $x_j$ and $x_i$, and is calculated as follows:

$$sgn(x_j - x_i) = \begin{cases} -1 & x_j - x_i < 0 \\ 0 & x_j - x_i = 0 \\ 1 & x_j - x_i > 0 \end{cases} \tag{7}$$

The length of the time series data was 19 years, and the statistic S followed a normal distribution. Therefore, the test statistic $Z$ was used to test for trend, and the test was taken at a significant level of $\alpha = 0.05$ and $Z_{1-\alpha} = Z_{0.975} = 1.96$ [47]. The Z test statistic was calculated as follows:

$$Z = \begin{cases} \frac{S+1}{\sqrt{VAR(S)}} & S < 0 \\ 0 & S = 0 \\ \frac{S-1}{\sqrt{VAR(S)}} & S > 0 \end{cases} \tag{8}$$

where $VAR(S)$ is the variance of the statistic $S$, calculated as follows:

$$VAR(S) = \frac{n(n-1)(2n+5) - \sum_{i=1}^{m} t_i(t_i - 1)(2t_i + 5)}{18} \tag{9}$$

where $n$ is the number of points in the sequence, $i$ is the number of repetitions, $m$ is the number of non-repetitive numbers, and ti is the number of repetitions in the $i$-th repetitive dataset.

### 2.4. Standardised Processing

To reduce errors in the evaluation results caused by differences in the nature, scale, and order of magnitude of the evaluation indicators in the multi-indicator evaluation system, it was necessary to standardise the raw data [48,49]. In this study, the minimum-maximum normalization method in ArcGIS10.8 was used to standardize GOSIF, GPP, and EVI data, as follows:

$$y_i = \frac{x_i - min(x_j)}{\max(x_j) - min(x_j)} \tag{10}$$

where $y_i$ is the result after normalisation of the *i*-th original data $x_i$ and $min(x_j)$ and $max(x_j)$ represent the minimum and maximum values of the original data, respectively.

### 2.5. Partial Correlation Analysis

Previous research has demonstrated that partial correlation analysis is an effective approach for examining the connection between vegetation phenological responses and environmental factors in arid regions [50,51]. Specifically, the onset of the phenological period appears to be more responsive to the mean temperature in March and April, precipitation in the preceding November, and the 12-month scale of the standardized precipitation evapotranspiration index (SPEI-12). In contrast, the conclusion of the phenological period is more sensitive to the mean temperature and precipitation in July and August, as well as SPEI-12 [52,53].

In this study, MATLAB R2021b software was used to select the mean temperature in March and April, the precipitation from November of the previous year, and SPEI-12 to analyse their relationships with the onset of the vegetation phenology. Similarly, the mean temperature, precipitation, and SPEI-12 in July and August were chosen as key climatic factors to examine their climate response in relation to the phenological conclusion [52,53].

For the bias correlation analysis between phenological and climatic factors, the other two factors were excluded, and the bias correlation coefficient was computed as follows:

$$R_{xy,z} = \frac{R_{xy} - R_{xz}r_{yz}}{(1 - R_{xz}^2)\left(1 - R_{yz}^2\right)} \tag{11}$$

where $R_{xy,z}$ is the correlation coefficient between $x$ and y after the control variable $z$, z is the other variable to be excluded, and $R_{xy}$, $R_{xz}$, and $R_{yz}$ represent the correlation coefficients of the variables $x$ and $y$, $x$ and $z$, and $y$ and $z$, respectively.

## 3. Results

### 3.1. Comparison of GOSIF- and MODIS-Based Phenology

3.1.1. Spatial Characteristics of GOSIF- and MODIS-Based Phenology

The distribution patterns of both the SOS_SIF and EOS_SIF are presented in Figure 2. The multi-year average SOS_SIF exhibited a "late–early–late" pattern from the southeast to the northwest. Within the designated research area, the multi-year average SOS_SIF for vegetation was 149 d, typically falling at the end of April. Vegetation SOS_SIF in this region was predominantly concentrated between 85 and 165 d, from late March to early June. More specifically, in Linze County of Gansu Province and the Junggar Basin area of the Tianshan Mountain Range in Xinjiang, located in the southeast portion of the study area, SOS_SIF was primarily concentrated between 85 and 150 d, from late March to late April. In contrast, in other regions, SOS_SIF was concentrated between 150 and 180 d, from early May to early June. Additionally, the study area exhibited noticeable spatial heterogeneity in vegetation phenology from north to south, with the northern region experiencing a later start to the growing season than the southern region.

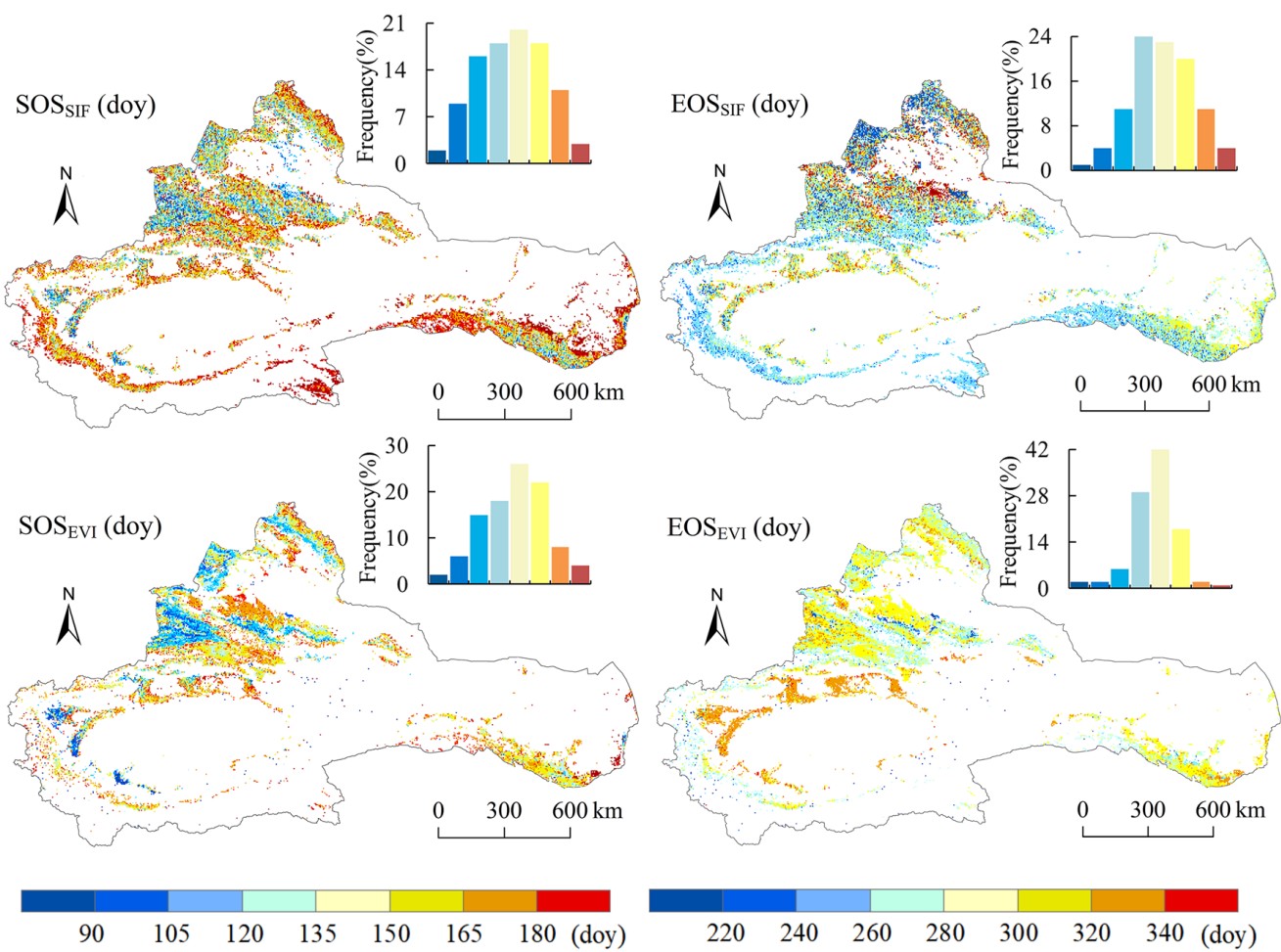

**Figure 2.** Spatial distribution map of vegetation phenology for SIF and MODIS.

$SOS_{EVI}$ was predominantly concentrated between 70 and 160 d, from mid-March to early June. The multi-year average of $SOS_{EVI}$ occurred 10 d earlier than that of $SOS_{SIF}$, and the spatial distribution patterns of $SOS_{EVI}$ and $SOS_{SIF}$ displayed similarities.

The multi-year average $EOS_{SIF}$ for the study area was 258 d, typically occurring in mid-September. In the southeast and northwest regions, $EOS_{SIF}$ was mainly concentrated between 260 and 310 d, from mid-September to early November. In contrast, in other regions, $EOS_{SIF}$ primarily occurred on day 310, in mid- to late-November. The multi-year average $EOS_{EVI}$ was predominantly concentrated between 260 and 320 d, from late September through late November, and it was on average 25 d later than $EOS_{SIF}$. The northern part of the study area saw an earlier conclusion to the growing season in contrast to the southern region. Relative to $EOS_{EVI}$, the spatial distribution of $EOS_{SIF}$ exhibited less heterogeneity and greater uniformity.

Figure 3 illustrates the month-by-month time series data for the mean values of GOSIF, EVI, and GPP in the study area from 2001 to 2019. The curves of these variables display a cyclical single-peak shape, portraying the dynamic attributes of the arid zone's vegetation growth cycle. The curves of GOSIF, EVI, and GPP all exhibit a cyclical single-peak shape, which corresponds to the typical pattern of vegetation growth cycles in arid regions. These curves exhibit a rapid ascent towards the end of April. The GOSIF curve peaked in mid-June during the growing season, followed by a swift decline. The EVI curve lagged behind the GOSIF curve but eventually coincided in mid-July. The EVI time series curve of MODIS rose earlier than the GPP curve. This finding suggests that the growing season duration derived from vegetation index data exceeded that determined using SIF and GPP.

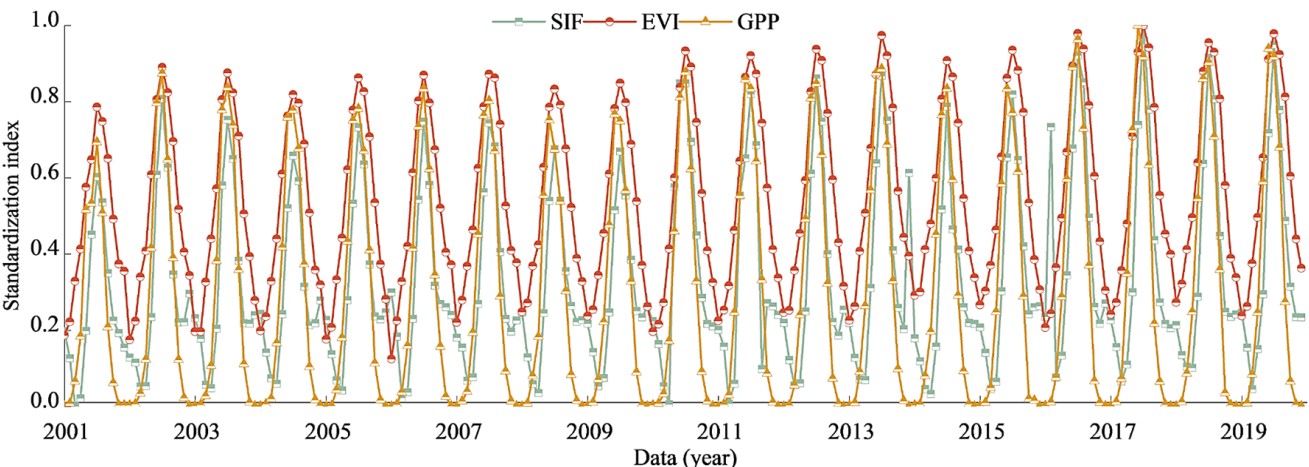

**Figure 3.** Time series curves of month-by-month SIF, EVI, and GPP averages from 2001 to 2019.

3.1.2. Temporal Trends between GOSIF- and MODIS-Based Phenology

Figure 4 presents the results of Sen trend analysis for vegetation $SOS_{SIF}$ and $EOS_{SIF}$ in the study area from 2001 to 2019. Both indicators demonstrated a fluctuating downward trend, with trend degrees of −0.22 and −0.49 d/yr, respectively, signifying an overall advancement in the start and end periods of vegetation growth. On the other hand, the overall $SOS_{EVI}$ and $EOS_{EVI}$ in the study area from 2001 to 2019 displayed a fluctuating upward trend. This indicates a delayed trend in the start and end periods of vegetation growth, with delayed trend degrees of 0.11 and 0.48 d/yr, respectively.

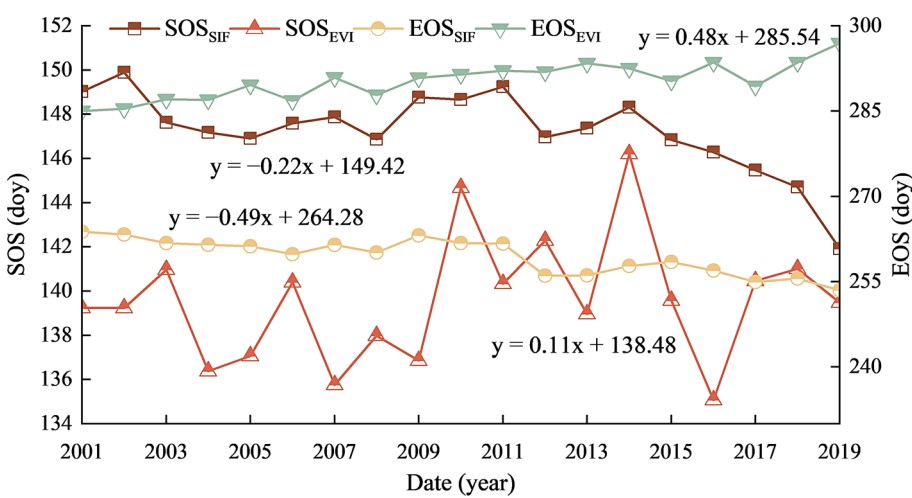

**Figure 4.** Time series of SEN trend of vegetation phenology in the arid zone.

Figure 5 presents the MK test results for the trend analysis of vegetation phenology from 2001 to 2019. Over 27.81% of the $SOS_{SIF}$ image elements displayed an advancing trend, with 9.35% being statistically significant. The significant image elements were primarily located in the Tianshan Mountains and Qilian Mountains. Similarly, more than 33.16% of the $EOS_{SIF}$ image elements exhibited an advancing trend, with 7.29% being statistically significant. The significant image elements were mainly distributed in the Tian Shan, Altay Shan, Qilian Shan, and Tarim Basins.

In contrast to the SIF climatic trend, 25.79% of the image elements in the vegetation climatic $SOS_{EVI}$ showed a delayed trend, with 6.96% being statistically significant. For $EOS_{EVI}$, 46.36% of the image elements displayed delayed trends, with 18.93% being statistically significant. The significant image elements were primarily situated in the Qilian Mountains and Altai Mountains.

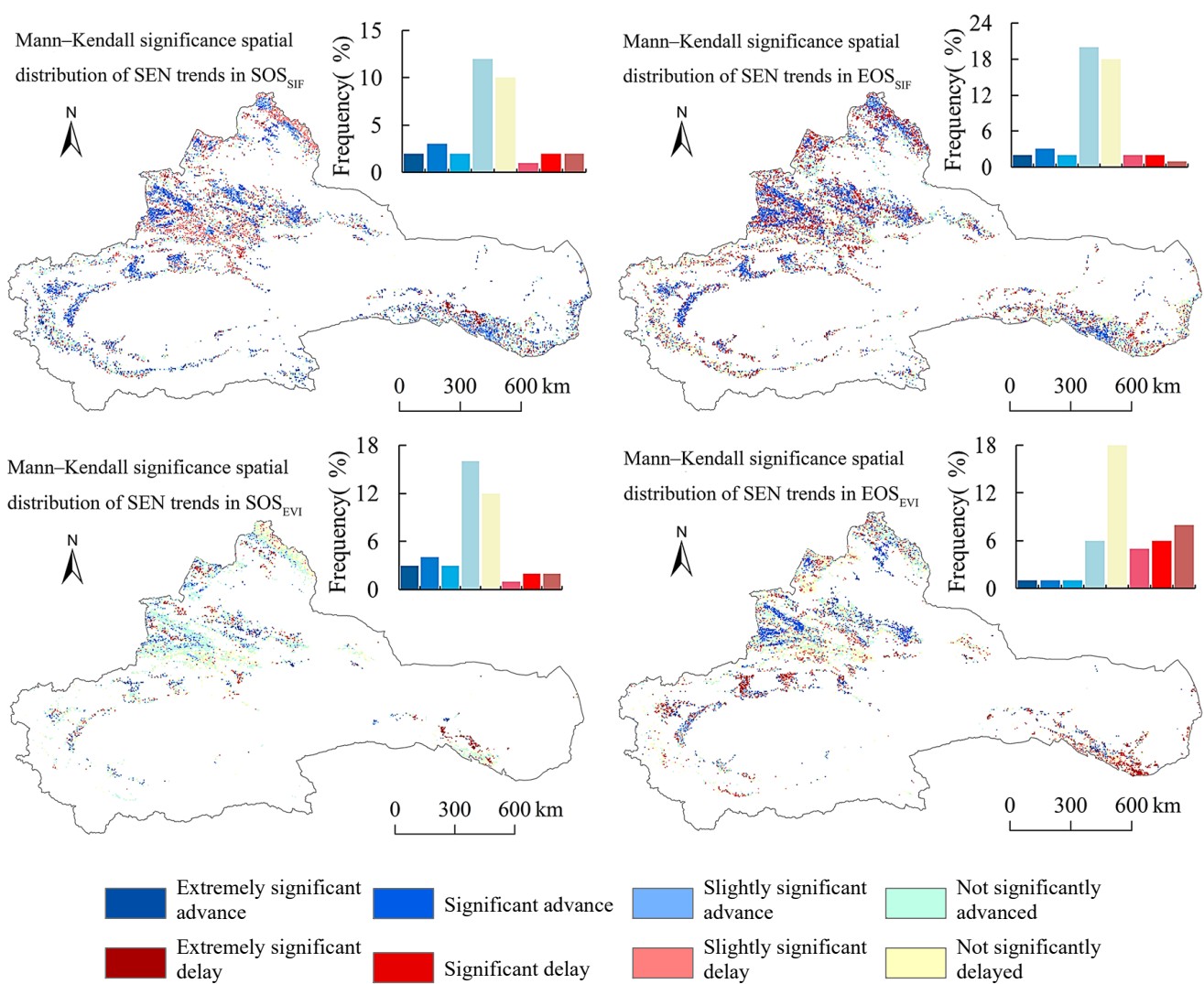

**Figure 5.** Spatial characteristics of Mann–Kendall significance test for SEN trend of vegetation phenology in the arid zone.

### 3.2. Sensitivity of Vegetation Phenology to Environmental Factors

3.2.1. Sensitivity of GOSIF- and MODIS-Based Phenology to Hydrothermal Changes

The results of the bias correlation analysis between the mean temperature in March–April and the precipitation in November of the previous year with corresponding climatic parameters for the same years revealed several insights (Figure 6). The average annual mean temperature from March to April exhibited a negative correlation with both $SOS_{SIF}$ and $SOS_{EVI}$. In other words, as the temperature increased, the beginning of the growing season advanced. Specifically, 64.99% of $SOS_{SIF}$ pixels and 62.86% of $SOS_{EVI}$ pixels in the study area were negatively correlated with temperature, with 7.10% and 5.29% of pixels showing a significant negative correlation. These significant correlations were primarily observed within the Tianshan region's northern and southern slopes, northern Altai, and western Qilian Mountains. Additionally, the average precipitation in November of the previous year showed a negative correlation with both $SOS_{SIF}$ and $SOS_{EVI}$ in 59.65% and 58.03% of the pixels, respectively. These negative correlations were primarily concentrated in the Tarim Basin, the eastern Altai Mountains, and the western Qilian Mountains.

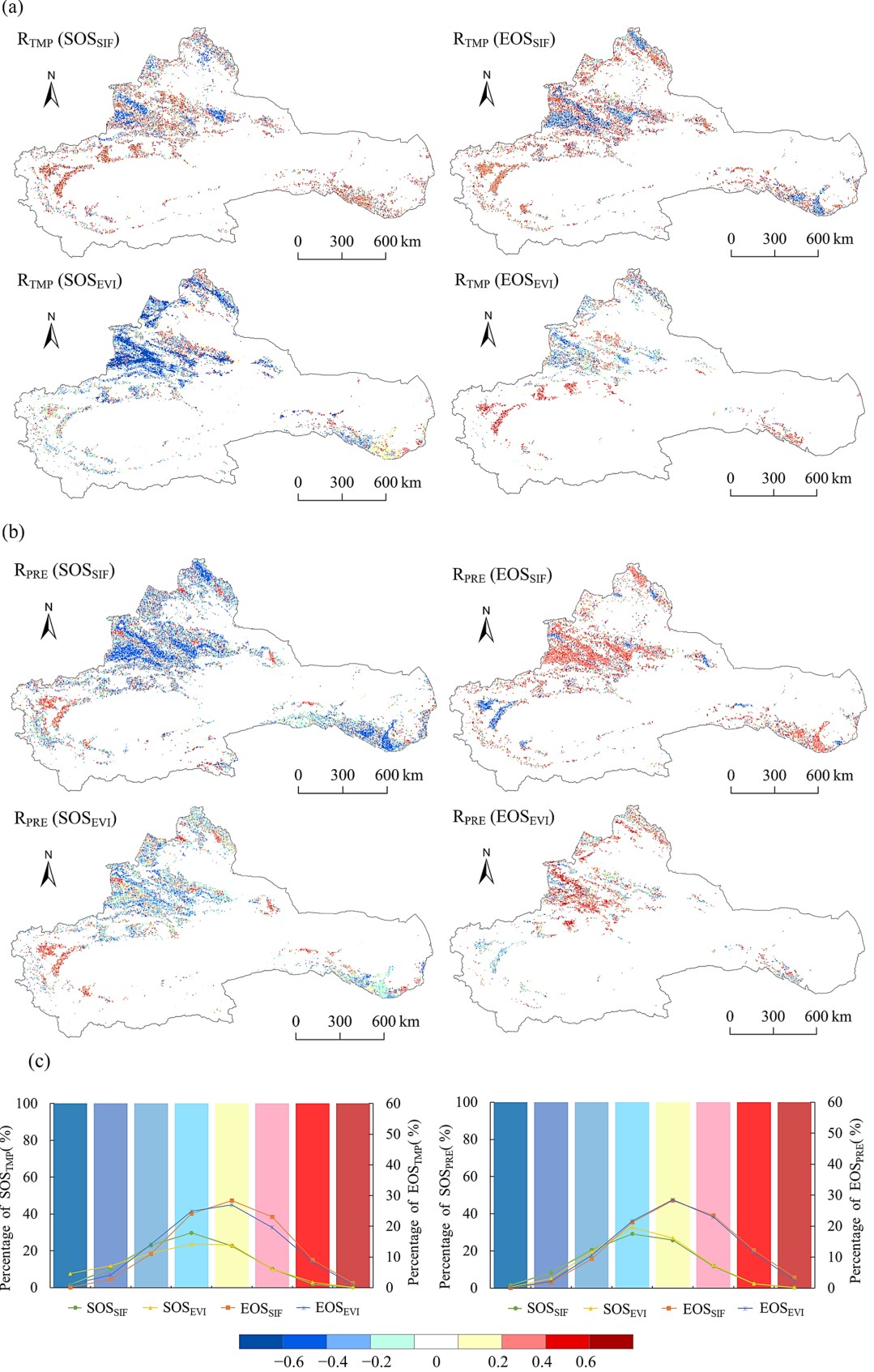

**Figure 6.** (**a**) Spatial representation of the bias correlation between vegetation phenology and temperature. (**b**) Spatial depiction of the bias correlation between vegetation phenology and precipitation. (**c**) Frequency plot of the bias correlation coefficient among vegetation phenology, temperature, and precipitation.

The impact of climatic factors on the termination of vegetation growth in the study area is more intricate. As indicated in Figure 6, 61.97% and 55.82% of the image elements of EOS$_{SIF}$ and EOS$_{EVI}$ exhibited positive correlations with the multi-year mean temperature in July and August. Additionally, 5.29% and 7.16% of the image elements had statistically significant positive correlations (Figure 7). These findings suggest that an increase in temperature leads to a delay at the end of the vegetation growth period. EOS$_{SIF}$ and EOS$_{EVI}$ also displayed positive correlations with 67.42% and 65.41% of the image elements with multi-year average July and August precipitation, respectively. Moreover, 7.79% and 9.11% of the image elements exhibited significant positive correlations. These findings suggest that greater precipitation is linked to a prolonged vegetation growth season. The significant image elements were primarily located in the Altay, Tianshan, and western Qilian Mountains.

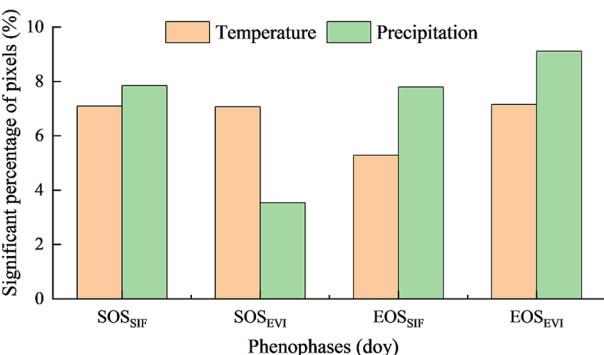

**Figure 7.** Frequency plot of significantly biased correlation of vegetation phenology with temperature and precipitation.

### 3.2.2. Sensitivity of GOSIF- and MODIS-Based Phenology to SPEI

When precipitation and temperature were constant, it was observed that in the study area, the proportion of image elements with a negative correlation between SOS$_{SIF}$ and SOS$_{EVI}$ and the annual mean SPEI was 59.65% and 58.03%, respectively (Figure 8). This means that, overall, with a decrease in the degree of drought, the beginning of the vegetation growth period has advanced. Of these correlations, 8.52% and 5.25% of the pixels displayed statistically significant negative correlations, primarily located in the eastern Altai Mountains, western Tianshan Mountains, Tarim Basin, and western Qilian Mountains.

Conversely, the proportions of EOS$_{SIF}$ and EOS$_{EVI}$ were positively correlated with the annual average drought degrees of 67.42% and 65.41%, respectively. In general, these findings suggest that the end of the vegetation growth period was delayed by a decrease in drought severity. Statistically significant positive correlations were observed in 6.66% and 3.92% of the pixels, mainly found on the southern slopes of the western Altai Mountains, Tianshan Mountains, and the Tarim Basin.

In summary, in comparison to SOS$_{EVI}$ and EOS$_{EVI}$, there were more image elements with significant correlations between vegetation SOS$_{SIF}$ and EOS$_{SIF}$ and SPEI in the study area. This indicates that vegetation SOS$_{SIF}$ and EOS$_{SIF}$ in arid regions exhibited greater sensitivity to SPEI.

### 3.2.3. Phenological Response to Environmental Factors across Diverse Vegetation Types

As depicted in Figure 9, the climatic responses of the three primary vegetation types (forest, grassland, and cropland) in the study area to temperature, precipitation, and SPEI extracted from GOSIF and MODIS data were consistent across the study area. However, GOSIF-extracted vegetation demonstrated greater sensitivity to climate in comparison to MODIS-extracted vegetation.

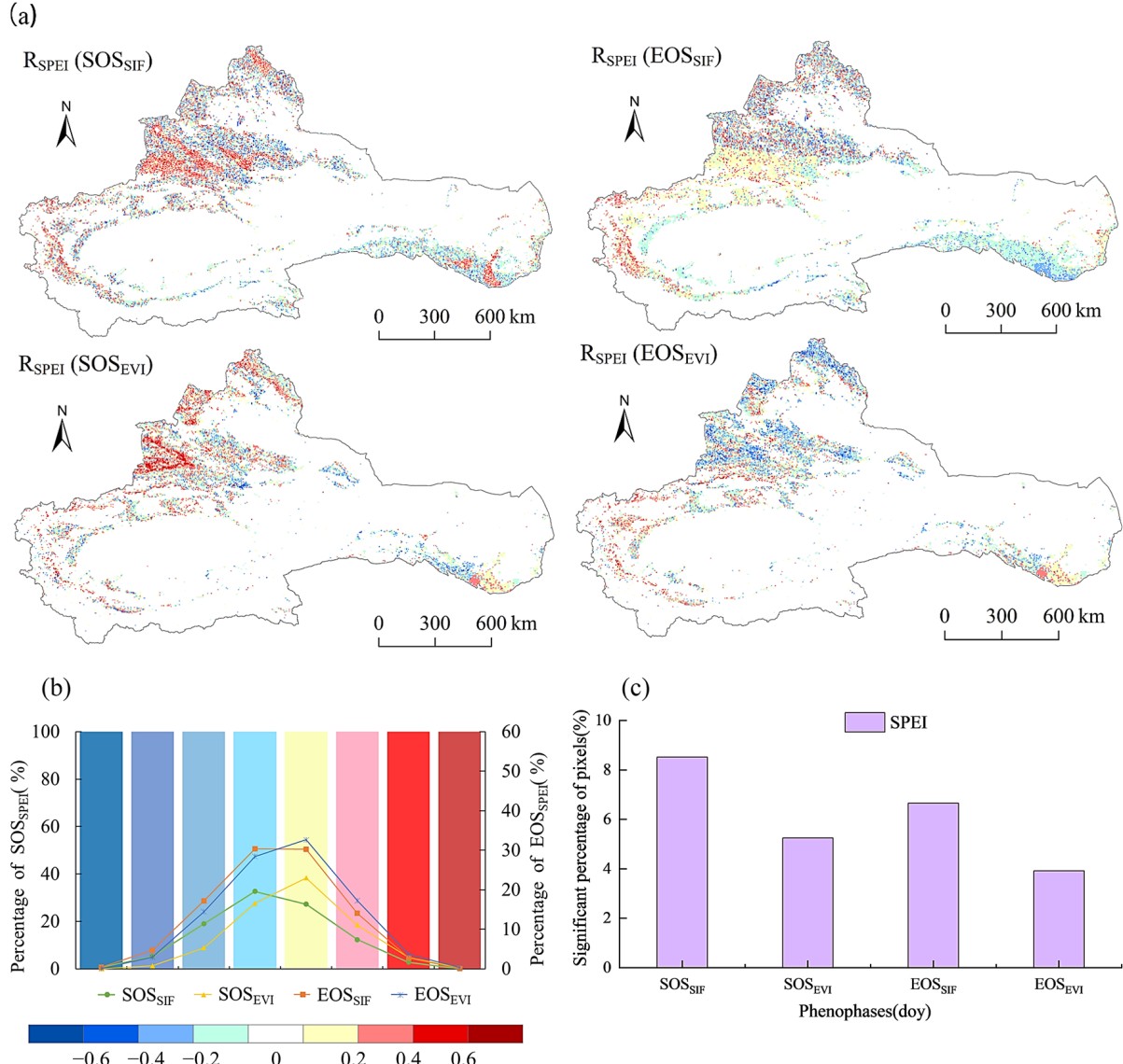

**Figure 8.** Bias correlation analysis between SIF- and MODIS-based phenology with SPEI changes in the arid zone. (**a**) Spatial pattern of bias correlation coefficients between vegetation phenology and SPEI. (**b**) Frequency plot displaying bias correlation coefficients between vegetation phenology and SPEI. (**c**) Frequency distribution of image elements with a significant bias correlation between vegetation phenology and SPEI.

For GOSIF-extracted forest growth, the bias correlation coefficients with climate factors were higher than those for MODIS-extracted forest growth. The difference between the bias correlation coefficients of GOSIF-extracted forest growth and climate, compared to MODIS forest growth and temperature, was most pronounced, with a difference of 0.063. This indicates that GOSIF-extracted forest growth was highly sensitive to all three climatic indicators.

In the case of croplands, the bias correlation coefficients of $SOS_{EVI}$ with temperature and SPEI were higher than those of $SOS_{SIF}$. Similarly, the bias correlation coefficient of $EOS_{EVI}$ with grassland temperature was greater than that of $EOS_{SIF}$, with a difference of 0.016. Furthermore, the bias correlation coefficient of $SOS_{SIF}$ with cropland precipitation was larger than that of $SOS_{EVI}$, with a difference of 0.027. This means that the $SOS_{SIF}$ of the croplands was more sensitive to precipitation than the $SOS_{EVI}$.

|  |  | Temperature | Precipitation | SPEI |
|---|---|---|---|---|
| SOS$_{SIF}$ | Forest |  | −0.616 | 0.526 |
|  | Grassland | −0.640 | 0.595 | 0.479 |
|  | Agricultural land | −0.526 | 0.582 | 0.447 |
| EOS$_{SIF}$ | Forest | 0.618 | 0.609 | −0.533 |
|  | Grassland | −0.561 | 0.586 | 0.565 |
|  | Agricultural land | 0.529 | 0.558 | 0.575 |
| SOS$_{EVI}$ | Forest | −0.618 | −0.557 | 0.522 |
|  | Grassland | −0.626 | 0.583 | 0.453 |
|  | Agricultural land | −0.534 | 0.555 | 0.487 |
| EOS$_{EVI}$ | Forest | 0.555 | 0.572 | −0.532 |
|  | Grassland | −0.600 | 0.577 | 0.556 |
|  | Agricultural land | 0.513 | 0.539 | 0.559 |

Partial correlation coefficient

**Figure 9.** Partial correlation coefficients of phenological response to climate for different vegetation types. The orange, green and mauve colours in the figure indicate the correlation of phenology with temperature, precipitation and SPEI, respectively, for different vegetation types.

### 3.3. Verification of Phenological Results

As depicted in Figure 10, Field-measured climate data from the Linze, Fukang, and Celle stations was used to validate and compare the vegetation estimated by the GOSIF and MODIS climate products. The validation results of the three stations in Figure 10 show that there was an overestimation of SOS$_{SIF}$ and an underestimation of SOS$_{EVI}$ compared with the observed data. However, SOS$_{SIF}$ was closer to the measured data. The EOS$_{SIF}$ and EOS$_{EVI}$ curves are distributed on both sides of the measured data curves. The SOS$_{SIF}$ curves show a later time than that of SOS$_{EVI}$, while the EOS$_{SIF}$ curves show a later time than that of EOS$_{EVI}$.

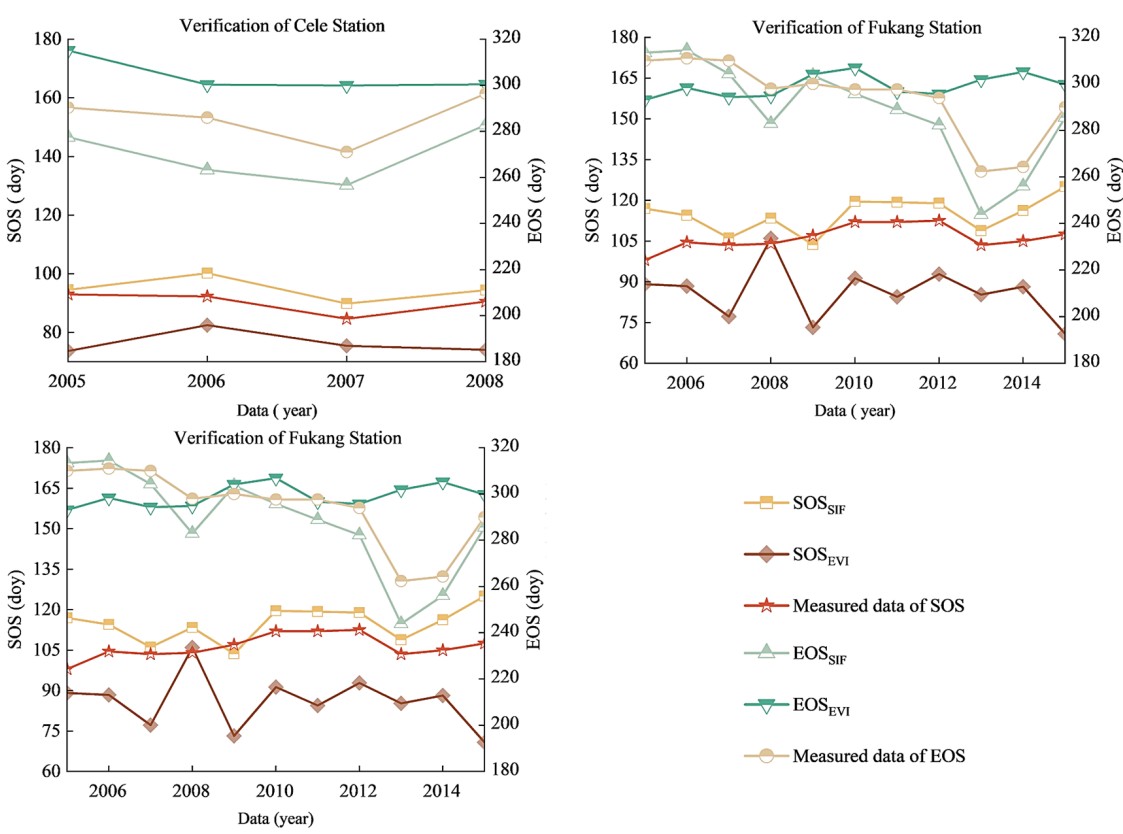

**Figure 10.** Validation of physical weather results based on site data.

## 4. Discussion

### 4.1. Temporal and Spatial Differences in GOSIF- and EVI-Based Phenology

This discrepancy can be attributed to the fact that the response to precipitation primarily dominates during the growth-opening period of croplands in the study area [54]. In contrast, the growth-ending period of grassland is influenced by a combination of various climatic factors, including precipitation and SPEI, and the response to precipitation is more sensitive during the growth-ending period of grassland [54,55]. Therefore, vegetation phenology data extracted using GOSIF can enhance the accuracy of phenological responses to climate change.

Figure 3 is consistent with prior research findings [56,57]. As SIF is the fluorescence signal emitted by plant leaves during photosynthesis, EVI is mainly used to reflect factors that include vegetation cover. This timing difference can be attributed to the fact that in the early stages of the growing season, the ecosystem is still in the carbon source stage, with plants beginning to respire before the leaf spreading stage. Photosynthesis of vegetation leaves occurs later than ecosystem respiration, causing the extraction of the beginning of the growing season based on vegetation index data to be earlier than that based on GOSIF data [58,59]. Similarly, at the end of the growing season, the degradation of chlorophyll in vegetation leaves is a slower process than the decline in photosynthesis, resulting in a lag in the extraction of the end of the growing season based on EVI data compared to that based on GPP data [60].

Figure 4 depicts the spatial trends of the mean values of $SOS_{SIF}$ and $SOS_{EVI}$ of the multi-year vegetation. The spatial distribution was characterised by a gradual advance and then a delay from southeast to northwest. The spatial trends of vegetation $EOS_{SIF}$ and $EOS_{EVI}$ were mainly characterised by advancement, whereas $SOS_{SIF}$ was delayed. The reason for the discrepancies in fall phenology could be attributed to the degradation of chlorophyll in leaves and the appearance of other pigments, such as carotenoids and anthocyanins, leading to changes in the reflectance of vegetation in visible and near-infrared spectra and impacting the accuracy of EVI-based fall phenology, whereas SIF-extracted fall phenology effectively avoids such spectral interference [5,45].

In addition, some studies have shown that the vegetation phenology reflected by SIF is more consistent with the wind and wave areas of the flux tower [61]. The $SOS_{SIF}$ curves are shown earlier than those in $EOS_{EVI}$. This difference may be because photosynthesis starts after plant leaf unfolding and stops before leaf senescence. The similarities and differences between the phenological periods extracted from GOSIF and MODIS EVI contribute to the understanding of the mechanisms that drive ecosystem phenological periods and their carbon cycling [28]. In addition, the differences between $EOS_{SIF}$ and $EOS_{EVI}$ indicate the complexity of the end period of vegetation growth (i.e., fall phenology). Previous findings have shown that autumn warming leads to increased carbon loss due to enhanced respiration [62]. As shown in Figure 10, the fall phenology extracted based on GOSIF ended earlier than that based on MODIS EVI. This is likely because vegetation may have stopped absorbing photosynthetic carbon in late fall but still maintained the greenness of the leaves [63]. This further elucidates why increased autumn warming could result in a net loss of ecosystem carbon. These findings provide important insights into vegetation phenology and reveal the influence of environmental factors on phenological periods and the mechanisms by which they affect carbon cycling. However, further studies are needed to validate these observations and probe the response mechanisms of SIF and EVI to phenology under different environmental conditions.

### 4.2. Uncertainty Analysis

This study considered two main aspects of uncertainty. First, there is a scale effect in remote sensing of vegetation phenology. The MODIS phenology product possesses a 500 m spatial resolution, and GOSIF has a spatial resolution of $0.05° \times 0.05°$. Regional differences in phenology estimated from remote sensing data with different spatial resolutions have been analysed [64]. In terms of temporal resolution, the reliability of sky-scale remote

sensing data is low because of the influence of the atmosphere and sensors. Furthermore, reconstruction methods for time series remote sensing data can introduce errors in the estimation of phenology. The S–G filtering method used in this study has a better effect on noise removal [65,66]. However, owing to the inability of the time series data curves of GOSIF and MODIS EVI to completely overlap, the growth cycle is not completely symmetrical, resulting in an algorithm for extracting the beginning of the vegetation growth period that is unable to extract the end of the growth period well [41].

Furthermore, while SIF is related to photosynthetic activity, photosynthesis in plants is a multifaceted physiological process affected by diverse environmental factors. Thus, accurately interpreting plant physiological states from SIF data may require the consideration of multiple factors to better understand the underlying processes [67]. This paper compares the applicability of SIF and EVI in estimating phenology in arid zones and their sensitivity to climate, so as to find the most suitable remote sensing index for estimating vegetation phenology in arid zones, and also provides a theoretical basis for responding to climate change by vegetation phenology in arid zones around the world. However, the accuracy of the results in this study was affected by the spatial resolution of the SIF and EVI. Therefore, the accuracy can be further improved using the following two methods. The first method integrates many types of new high-resolution remote sensing indices to improve the climate monitoring ability of SIF, such as VPP, PPI, and NDGI. Secondly, it is also possible to combine meteorological factors to calibrate the critical period of phenology for different vegetation types based on existing remote sensing estimations of phenology to improve the precision of phenology monitoring in SIF [68,69].

## 5. Conclusions

In this study, a comparative analysis was performed to examine how climatic factors affected vegetation phenology in the arid region of Northwest China from 2001 to 2019, utilising data from GOSIF and MODIS EVI. The key findings of the study are as follows:

(1) The overall $SOS_{SIF}$ of the perennial vegetation in the study area was later than the $SOS_{EVI}$, whereas the overall $EOS_{SIF}$ was earlier than the $EOS_{EVI}$. Validation results indicated that SIF-based phenology estimations were more consistent with ground-truth data than those derived from MODIS EVI. The beginning and ending phases of vegetation phenology growth exhibited similar spatial patterns, but the ending phase showed more significant spatial heterogeneity compared to the beginning phase. The spatial distributions of the change trends of $SOS_{SIF}$ and $SOS_{EVI}$ were relatively consistent. However, the spatial trends of $EOS_{SIF}$ and $EOS_{EVI}$ varied; $EOS_{SIF}$ mainly exhibited a trend of advancement, while $SOS_{SIF}$ exhibited a trend of delay in $SOS_{SIF}$.

(2) The vegetation phenology extracted from GOSIF was more sensitive to temperature, precipitation, and SPEI compared to that derived from MODIS EVI. Temperature, precipitation, and SPEI were negatively correlated with the initiation of the vegetation growth period, while temperature was negatively correlated with the end of the growth period. Precipitation and SPEI were positively correlated with the end of the growth period. In terms of spatial distribution, vegetation phenology showed a higher level of sensitivity to climate factors in specific regions, including the Altay Mountains, Tianshan Mountains, western Qilian Mountains, and Tarim Basin.

(3) The vegetation phenology of forests, grasslands, and croplands extracted using GOSIF exhibited higher sensitivity to temperature, precipitation, and SPEI compared to those derived from MODIS EVI. Specifically, croplands exhibited greater sensitivity to precipitation, and the fall phenology of grasslands was primarily influenced by precipitation and SPEI. These findings indicate that employing SIF extraction to investigate the response of vegetation phenology to climate change in arid regions can yield more scientifically meaningful insights.

In summary, the study demonstrates the effectiveness of using SIF data (GOSIF) for assessing vegetation phenology and how it reacts to climatic factors in arid regions, highlighting its advantages in terms of sensitivity and accuracy compared to MODIS EVI.

These findings emphasise the significance of considering SIF extraction when conducting future research on vegetation phenology in arid zones.

**Author Contributions:** Conceptualization, Z.C. and M.Z.; methodology, Z.C.; software, Z.C.; validation, J.K., C.X. and S.Y.; formal analysis, M.Z.; investigation, Z.C.; resources, Z.C.; data curation, J.K.; writing—original draft preparation, Z.C.; writing—review and editing, M.Z.; visualization, Z.C.; supervision, M.Z.; project administration, M.Z.; funding acquisition, M.Z. All authors have read and agreed to the published version of the manuscript.

**Funding:** This research was funded by the National Natural Science Foundation of China, grant number (NO.42261013), Natural Science Foundation of Xinjiang Uygur Autonomous Region grant number (NO.2023D01A49).

**Data Availability Statement:** The code and data used in this work are available from the corresponding author upon reasonable request.

**Conflicts of Interest:** The authors declare no conflict of interest.

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
