# Peer review of "Phenology of Vegetation in Arid Northwest China Based on Sun-Induced Chlorophyll Fluorescence"

_forests, doi:10.3390/f14122310_

Round 1

Reviewer 1 Report

Comments and Suggestions for Authors

Dear Authors,

The paper is devoted to mapping vegetation phenology in arid Northwest China using remote sensing data. Please find my recommendations for improving the manuscript.

1. Please correct title of the manuscript with no present simple statement
2. Please improve the abstract higlighiting novel approach applied in your research study. What is new, innovative comparing to other previous studies?
3. I realize the research work including coarse spatial resolution MODIS EVI and GOSIF on mapping vegetation pehnology (SOS, EOS etc) is repetitive. Please rewrite Introduction underlying your approach: what are the results expected you wish to publish? Why EVI only you utilized, if there are many sophisticated vegetation indices used for phenology monitoring such as Plant Phenology Index (PPI), VPP Copernicus (https://custom-scripts.sentinel-hub.com/custom-scripts/copernicus_services/hrvpp/)?

4. Please answer the question what new is in your research study in relation to similar studies such as listed below:
https://www.sciencedirect.com/science/article/pii/S0303243420300209?via%3Dihub

https://ieeexplore.ieee.org/document/8518142

https://www.mdpi.com/2072-4292/10/1/122

5. Please sharpen your all figures with maps and profiles. The figures quality is not to be accepted.

6. Please add into Introduction a paragraph on machine learning approaches for mapping vegetation growth and state (phenology as well) to be used for supporting coarsed spatial resolution MODIS and GOSIF:

https://www.mdpi.com/2072-4292/15/9/2392

7. In the Discussion I would ask you for adding paragraph on applicability many other vegetation indices to be utilized for spatiotemporal chlorophyll fluorescence estimations allowing mapping phenology at finer spatial resolution. Please cite the following works:

https://www.sciencedirect.com/science/article/abs/pii/S016819232200020X
https://www.mdpi.com/2072-4292/15/9/2413

8. In my opinion the results you achieved and the title indicate very repetitive statements and conclusions because SIF is well known of direct relation to photosynthetic activity while EVI on general vegetation condition. It is commonly known SIF is much more precise for evaluating phenology trajectories, however SIF limited spatial resolution brings on investigating other vegetation indices for improving vegetation monitoring with high spatial resolution.

Author Response

we are very grateful to all the teachers and experts in the editorial department for their valuable comments on our paper. We have carefully read and scrutinized all the comments and suggestions, and made very serious thinking and modification for each issue, and the revisions were highlighted in yellow in the appropriate places in the article, and please see the attachment.

Reviewer 2 Report

Comments and Suggestions for Authors

The article is devoted to the comparison of Solar–Induced Fluorescence measurements from OCO-2 and MODIS satellite data with EVI and meteorological data for arid Northwest China territory. The authors collected data from 2001 to 2019, processed them with filtering methods, and examined their trends. Overall, the manuscript presents a good study on the relevant topic of vegetation properties studying from remote sensing data. I have a few remarks, which are more related to the introduction and design of the paper.

1.      The title of the manuscript is presented as a statement, a conclusion, rather than a research topic. In my opinion, the conclusion in the title does not encourage the reader to read the paper.

2.      There is too little written in the Introduction about the SIF. Please add some details - what are the features of its measurement, what disadvantages are known.

3.      In Section 2.2, you write you will compare three indices – SIF, EVI, GPP. But why are these particular indices taken, what is the evidence of their similarity? It should be in the Introduction.

4.      As for GPP, you only mention the data source for it and then it only appears in the Discussion. If you have used this index in your work, add the source information about it in the Introduction, describe its processing in the Methods and show the Results of processing. Otherwise, the Methods now only describe the methodology of processing the SIF.

5.      The Discussion section contains specific results of calculations not shown in the Results section. I recommend everything related to Figures 8-10 should be moved to the Results section, and only the reasoning (why such results were obtained, what are the limitations, etc.) should remain in the Discussion section.

6.      Section 2.2.3 describes the MCD12Q2 product, but I couldn't find the start and end of growth parameters in the description on the NASA website. Are these the Greenup and Dormancy parameters? Please specify the exact name of the parameters from MCD12Q2.

7.      You specified the TIMESAT3.3 software for the S-G filter, but what software was used to do the rest of the data processing?

8.      There is no north arrow in Figure 1.

9.      SOSSIF, etc. are written merged in some places rather than with a subscript SIF.

10.  Line 221: typo in the SIF subscript: “radio” instead of “ratio”.

11.  Lines 225-226: The sentence is repeated twice “EOSSIF was defined as the DOY…”

12.  Line 230: incorrect numbering of the subsection 3.1.3 instead of 2.1.3

13.  All the map images are too blurry.

14.  Figures 2 and 3: abbreviations “DOY” are in small letters “doy” instead of capital letters.

Author Response

(The authors gave the same response as above.)

Round 2

Reviewer 1 Report

Comments and Suggestions for Authors

Dear Authors,

I accept in present form.

Kindest regards

Reviewer 2 Report

Comments and Suggestions for Authors

The authors have corrected all my remarks and responded to all comments. I think that the paper can be accepted in present form